# Beyond Oncology: Question Prompt Lists in Healthcare—A Scoping Review Protocol

**DOI:** 10.3390/mps3010009

**Published:** 2020-01-16

**Authors:** Matthias Lukasczik, Christian Gerlich, Hans Dieter Wolf, Heiner Vogel

**Affiliations:** Section of Medical Psychology and Psychotherapy, Center of Mental Health, Würzburg University Hospital, D-97070 Würzburg, Germany; christian.gerlich@uni-wuerzburg.de (C.G.); hans.wolf@uni-wuerzburg.de (H.D.W.); h.vogel@uni-wuerzburg.de (H.V.)

**Keywords:** question prompt list, patient participation, scoping review, decision support techniques, literature search, non-oncological

## Abstract

Question prompt lists (QPL) are an instrument to promote patient participation in medical encounters by providing a set of questions patients can use during consultations. QPL have predominantly been examined in oncology. Less is known about their use in other contexts. Therefore, we plan to conduct a scoping review to provide an overview of the fields of healthcare in which QPL have been developed and evaluated. MEDLINE/PUBMED, PSYCINFO, PSYNDEX, WEB OF SCIENCE, and CINAHL will be systematically searched. Primary studies from different healthcare contexts that address the following participants/target groups will be included: persons with an acute, chronic, or recurring health condition other than cancer; healthy persons in non-oncological primary preventive measures. There will be no restrictions in terms of study design, sample size, or outcomes. However, only published studies will be included. Studies that were published in English and German between 1990 and 2019 will be examined. Two independent reviewers will apply defined inclusion/exclusion criteria and determine study eligibility in the review process guided by the PRISMA statement.

## 1. Introduction

Against the backdrop of the idea of patient-centered medicine [1], it is vital that patients are informed sufficiently and appropriately about their disease in order to achieve active participation in health-related decisions (e.g., specific treatment options and their potential benefits and risks), adherence with the treatment regimen, and self-management/disease management [2,3,4].

This is particularly relevant in chronic (and/or potentially life-threatening) diseases such as cancer where treatment-related decisions may have far-reaching consequences. For instance, research has shown high information needs in cancer patients, varying levels of satisfaction with the quality and amount of information received, and considerable unmet needs [5,6,7,8]. Factors such as age [9], type of cancer [7], or health literacy level [10] may moderate information needs and satisfaction with information.

Information needs are also prevalent in health conditions such as acute illness or injury [11,12,13] where it is equally important that patients have the opportunity to obtain information about the illness, its therapy, and other related aspects.

A wide range of instruments and approaches is used to promote higher levels of patient participation and empowerment in medical encounters [14]. These approaches address healthcare professionals (mainly physicians), patients, or both. Among them are printed or online information materials for patients [15], specific decision aids [16], communication skills trainings for physicians (and, to a lesser extent, other healthcare professionals) [17,18,19], and patient education programs, often in the context of chronic disease self-management [20,21,22].

Some of these methods have a more strictly defined purpose such as decision aids. They may also be embedded in a broader framework such as Shared Decision-Making (SDM) [23,24] that seeks to establish a viable physician–patient relationship, thus enabling patients to obtain (or ask for) relevant information. 

Among the methods that address patients directly and encourage them to actively gather information are communication skills trainings [25] and question prompts (or question prompt lists, QPL). The latter can be used as an inexpensive communication aid in medical/healthcare encounters. QPL are structured lists of questions provided to patients who can use these questions during a consultation [26,27,28]. Patients may also use subsets of QPL questions relevant to them or add or generate own questions [28]. The use of QPL aims at prompting physicians to provide relevant information about the respective illness, its treatment, or other aspects.

QPL have been developed and evaluated mainly in oncology [26]. Research has shown that their use is associated with more question asking by patients, higher levels of information satisfaction, and a reduction of unmet information needs [14,29,30,31,32,33]. Patients rate QPL as helpful [30,34,35,36,37,38] and are more satisfied with consultations in which QPL were used [39].

In contrast, QPL in non-oncological contexts are less common, with examples including family medicine, kidney disease, depression, and surgery [40,41,42,43]. One current review on the effects of QPL (in terms of total question asking, provision of information, knowledge recall, and patient satisfaction, among others) included both cancer and non-cancer settings [28].

There is, however, limited information on how and with what purpose QPL are used in varying non-oncological contexts. For instance, while the review conducted by Sansoni et al. [28] differentiated QPL by content area and type, it did not explicitly take into account the role of the particular health condition or context (e.g., family practice, medical rehabilitation) and its specifics.

Against this background, the planned scoping review presented in this protocol shall address the following question: In which areas of healthcare except oncology have question prompts been developed, evaluated, and implemented? The review shall provide insight into the contexts as well as the purpose/objectives of QPL (i.e., are they comparable to oncology, in what respect do they differ). The outline of this protocol is guided by the recommendations of Peters et al. [44].

## 2. Methods and Analysis

### 2.1. Objectives and Methodology

To describe the range in which QPL in fields other than oncology have been developed, evaluated, and implemented in terms of indications/medical specialties, outcomes (i.e., have outcome variables in terms of effects (or correlates) of QPL use have been measured, such as knowledge recall, satisfaction), objectives/intended purpose (e.g., as decision aid, as a tool to promote health-/disease-related knowledge), and implementation (if inferable from the extracted studies). In light of this purpose, it seemed appropriate to conduct a scoping review as this type of review (as distinguished from a systematic review) aims at identifying and synthesizing the scope and types of available evidence and mapping a research topic or concept, which is, in this case, the use of QPL in a non-oncological context [44,45].

### 2.2. Context/Setting

The review will focus on all non-cancer-related healthcare settings. This comprises primary/secondary prevention; primary care; medical rehabilitation; palliative care. Both inpatient and outpatient/ambulatory settings will be included.

### 2.3. Study Types; Inclusion and Exclusion Criteria

Primary studies that report on the development, formative and/or summative evaluation, and/or implementation of QPL will be included in the review. There will be no restrictions in terms of study design (i.e., inclusion of all types of studies, such as cohort studies, RCTs, case-control studies; both quantitative and qualitative designs, will be considered). However, all types of reviews (literature reviews, systematic reviews, meta-analyses) will be excluded.

No restrictions in terms of sample size or outcomes will be defined. However, only published studies will be included. We decided not to consider “gray” literature due to the intended focus on those sources that depict the development and evaluation of QPL in an evident research framework (i.e., publications in scientific journals). Admittedly, this may imply a certain loss/neglect of potentially relevant information.

### 2.4. Participants

Studies that address the following participants or target groups will be included: persons with an acute, chronic, or recurring health condition other than cancer; healthy persons in primary preventive measures (other than cancer-related interventions, such as mammography/cancer screenings, etc.); persons with initial non-oncological health problems or symptoms in secondary preventive measures.

No restrictions will be made with regard to participant age or sex. Not restricting the age range will render it possible to include studies in which QPL are developed for parents of children with a disease, for example.

### 2.5. Search Strategy

The search will be restricted to studies published in English and German between 1990 and 2019. In their review, Sansoni et al. [28] limited their search to the period from 2000 to 2015 but stated that they would also include older publications, if relevant, which was the case. Therefore, it seemed reasonable to define a priori a broader publication date range. Including German-language studies allows an estimate of the role of QPL in a specific national context (in this case, the German healthcare system and associated research). 

Search terms will include the keywords presented in Table 1. 

Search terms will also include Medical Subject Headings (MeSH) related to (1) patient participation, (2) physician–patient relations, (3) decision making, and (4) decision support techniques.

The following databases will be systematically searched: MEDLINE/PUBMED; PSYCINFO; PSYNDEX; WEB OF SCIENCE; CINAHL.

For each review step (see below), the following information will be documented: date searched, database searched, number of hits.

### 2.6. Data Extraction and Charting

The review decision process will be based on the following review steps whose results will be detailed in a flow chart guided by the PRISMA statement [44,46]: Identification of records through database searching; screening of records (based on title and abstract) after the removal of duplicates; full-text assessment of articles for eligibility; final selection of studies included in the review. The references of studies undergoing full-text assessment will be screened for additional publications (citation tracking) which will be subjected to screening and assessment for eligibility if applicable.

At each stage of the review process, two independent reviewers will be involved. In case there is disagreement among reviewers, a third independent reviewer will assess the respective publication. If no agreement is reached, the publication will be excluded from further assessment.

In those cases where no sufficient information can be extracted from a (full-text) publication, there will be an attempt to contact the study authors. After a time frame of 30 days has expired without a response, the respective publication will be excluded. 

Key characteristics of studies included as well as relevant information derived from them will be summarized in table form. 

Results will be summarized with reference to the review question, i.e., with regard to (a) indications/medical specialties; (b) outcomes and intended purpose; (c) routine implementation (if inferable from studies). A graphical summary of results will be charted.

## 3. Expected Results and Implications

There is evidence—mainly based on research in cancer care—that question prompt lists encourage patient participation in medical encounters. The review protocol shall initiate a scoping review that seeks to identify evidence on question prompt lists in non-cancer-related healthcare contexts. The planned review shall provide insight into the use of question prompt lists in various healthcare contexts, thus delivering a better appraisal of their utility as a tool to support patient participation.

## Figures and Tables

**Table 1 mps-03-00009-t001:** Keywords used as search terms.

Keywords in English	Keywords in German
“question prompt” “question prompts” “question prompt list” “question prompt instrument” “question prompt sheet” “question prompt tool” “patient questions” “asking questions” “question asking” “question asking tool” “question asking aid” “patient question asking” “patient question list” “patient question asking tool” “patient question asking aid” “patient question asking list” “prompt list” “prompt instrument” “prompt sheet” “prompt tool” “question aid” “patient question aid” “patient decision aid”	„Fragenliste“ „Patientenfrageliste“ „Gesprächshilfe“ “Gesprächsliste”

Note: It can be assumed that the English term “question prompt (list)” will also be used in German-language publications compared with a direct German translation of the term that would be rather cumbersome. Thus, there is not an equivalent for each English keyword.

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
