# Peer review of "Beyond Oncology: Question Prompt Lists in Healthcare—A Scoping Review Protocol"

_mps, 2020, doi:10.3390/mps3010009_

Round 1
Reviewer 1 Report
Thankyou for the opportunity to peer review this scoping review protocol. My comments are outlined below. The main concern relates to clarifying how the 'evaluation' and 'implementation' components of the review objective will be measured.
I encourage the authors to remain consistent in the terminology used to describe the context. Across the manuscript the terms 'fields', 'settings' and 'areas' seem to be interchangable
Line 48-54 - some more information about QPL would be useful here e.g. who develops them? Is the decision to use QPL up to healthcare professionals?
Line 62 -what did the review find for non-cancer settings?
Line 76 - 'implemented' was not mentioned in the review question outlined in the paragraph above
Line 76 - please clarify 'outcomes' - is this related to the evaluation part of the review question? It is unclear how the evalaution part of the obective will be measured
Line 78 - how will 'implementation' be measured? This is simliarly unclear at line 142 (degree of implementation)
Lines 84-6 - perhaps this section could be simplified or at least re-iterated by stating the protocol will focus on all non-cancer related healthcare settings?
Line 115 - it would be useful to provide a full search strategy including the date searched, database searched and number of hits. If this is not a requirement for the jounal it would help to indicate how search terms will be combined and what fields will be searched
Line 123 - Why has EMBASE not been considered?
Line 130 - please clarify what is meant by 'selected studies' e.g. those included in the review? those assessed at full text?
Author Response
We would like to express our gratitude to the reviewer for his thorough and useful suggestions regarding our manuscript. We have responded to the comments as outlined below.
I encourage the authors to remain consistent in the terminology used to describe the context. Across the manuscript the terms 'fields', 'settings' and 'areas' seem to be interchangable.We thank the reviewer for this valuable suggestion. Throughout the text, we have harmonized the terminology where necessary.
Line 48-54 - some more information about QPL would be useful here e.g. who develops them? Is the decision to use QPL up to healthcare professionals?To our knowledge, QPL are generally developed in the context of research projects/studies. We are not aware of systematic information on whether/to what extent QPL are being transferred into routine care. Please note that we have not included this statement in the revised document. We will be happy to do so if the reviewer thinks that this is reasonable.
Line 62 -what did the review find for non-cancer settings?The review results did not differentiate between specific settings or diseases (or cancer vs. non-cancer). At some point, it referred to one included study focusing on disease knowledge in cardiovascular patients. Please note that we have not included this statement in the revised document. We will be happy to do so if the reviewer thinks that this is reasonable.
Line 76 - 'implemented' was not mentioned in the review question outlined in the paragraph above.We thank the reviewer for pointing out to this discrepancy. We have revised the paragraph accordingly.
Line 76 - please clarify 'outcomes' - is this related to the evaluation part of the review question? It is unclear how the evalaution part of the obective will be measured.We have revised this passage. Provided that this information is available, we want to document whether effects or other outcome variables/correlates of the QPL and its use (i.e., “outcomes”) have been assessed/measured in the respective study.
Line 78 - how will 'implementation' be measured? This is simliarly unclear at line 142 (degree of implementation).Thank you for pointing out to this important aspect. We plan to extract information (provided it is available) from the studies on whether the QPL was implemented in routine care. This may be the case in terms of (qualitiative, quantitative) data, (unsystematic) observations or simply statements regarding implementation. Thus, we assume that data on implementation will be available in certain cases only. Please note that we have not included this statement in the revised document. We will be happy to do so if the reviewer thinks that this is reasonable.
Lines 84-6 - perhaps this section could be simplified or at least re-iterated by stating the protocol will focus on all non-cancer related healthcare settings?We have revised the first sencence. However, we would like to keep the list of settings for reasons of transparency.
Line 115 - it would be useful to provide a full search strategy including the date searched, database searched and number of hits. If this is not a requirement for the journal it would help to indicate how search terms will be combined and what fields will be searched.We have added the relevant information as suggested by the reviewer and would like to thank for this important suggestion.
Line 123 - Why has EMBASE not been considered?We presume there will be a high level of overlap/redundancy between EMBASE and MEDLINE (which is one of the databases to be used during the review process); therefore, we decided not to consider EMBASE.
Line 130 - please clarify what is meant by 'selected studies' e.g. those included in the review? those assessed at full text?We thank the reviewer for this helpful suggestion. Accordingly, we have revised this section.
Reviewer 2 Report
Thank you for the opportunity to review the manuscript;
I have the following comments for you to consider;
The section detailing inclusion and exclusion criteria needs to be divided into two subsections; types of studies and participants. It is unclear which type of participants you will be including. Regarding the type of studies; you need to mention if they are qual. or quant, reviews, reports, etc.. no need to mention sample size. Your search terms need to include terms relevant to your participants, concept and context. There is no data synthesis in a scoping review, just data charting, plesae review this section Plesae consider including a table detailing the results which is based on your PCC and study detailsAuthor Response
We would like to express our gratitude to the reviewer for his thorough and useful suggestions regarding our manuscript. We have responded to the comments as outlined below.
The section detailing inclusion and exclusion criteria needs to be divided into two subsections; types of studies and participants. It is unclear which type of participants you will be including. Regarding the type of studies; you need to mention if they are qual. or quant, reviews, reports, etc.. no need to mention sample size.We thank the reviewer for this suggestion. Accordingly, we have divided the abovementioned section into subsections “types of studies” and “participants”. From our point of view, the information addressed by the reviewer are already included in these subsections. Therefore, we would like to refer to the respective paragraphs in the sections now numbered as 2.3 and 2.4, respectively.
Your search terms need to include terms relevant to your participants, concept and context.We would like to thank the reviewer for this valuable suggestion. Our intention was to identify a broad range of QPL studies irrespective of setting, study type or participant group. We thought that this could be achieved best by not restricting the search using more specific terms (such as specific patient groups or contexts). Please note that we have not included this statement in the revised document. We will be happy to do so if the reviewer thinks that this is reasonable.
There is no data synthesis in a scoping review, just data charting, plesae review this section.We would like to thank the reviewer for pointing out to this aspect. We have adapted this section accordingly.
Plesae consider including a table detailing the results which is based on your PCC and study details.Since we are not sure what is meant by PCC, we would be grateful for a clarification by the reviewer. Currently, we have not included this aspect in the revised document.